# Microbial Detoxification of Sediments Underpins Persistence of *Zostera marina* Meadows

**DOI:** 10.3390/ijms25105442

**Published:** 2024-05-16

**Authors:** Yuki Nakashima, Takumi Sonobe, Masashi Hanada, Goushi Kitano, Yoshimitsu Sonoyama, Katsumi Iwai, Takashi Kimura, Masataka Kusube

**Affiliations:** 1Graduate School of Frontier Biosciences, Osaka University, Suita 565-0871, Osaka, Japan; u021527c@ecs.osaka-u.ac.jp; 2Advanced Engineering Faculty, National Institute of Technology, Wakayama College, Gobo 644-0023, Wakayama, Japan; 3Promotion of Technical Support, National Institute of Technology, Wakayama College, Gobo 644-0023, Wakayama, Japan; 4Agri-Light Lab. Inc., Minamata 867-0068, Kumamoto, Japan; 5Study Team for Creation of Waterfront, Yokohama 220-0023, Kanagawa, Japan; 6Department of Applied Chemistry and Biochemistry, National Institute of Technology, Wakayama College, Gobo 644-0023, Wakayama, Japan

**Keywords:** *Zostera marina*, eelgrass meadows, blue *carbon*, H_2_S detoxification, sulfur-oxidizing bacteria, sulfur-reducing bacteria, *Chromatiales*

## Abstract

Eelgrass meadows have attracted much attention not only for their ability to maintain marine ecosystems as feeding grounds for marine organisms but also for their potential to store atmospheric and dissolved CO_2_ as blue carbon. This study comprehensively evaluated the bacterial and chemical data obtained from eelgrass sediments of different scales along the Japanese coast to investigate the effect on the acclimatization of eelgrass. Regardless of the eelgrass habitat, approximately 1% *Anaerolineales*, *Babeliales*, *Cytophagales*, and *Phycisphaerales* was present in the bottom sediment. Sulfate-reducing bacteria (SRB) were present at 3.69% in eelgrass sediment compared to 1.70% in bare sediment. Sulfur-oxidizing bacteria (SOB) were present at 2.81% and 1.10% in the eelgrass and bare sediment, respectively. Bacterial composition analysis and linear discriminant analysis revealed that SOB detoxified H_2_S in the eelgrass meadows and that the larger-scale eelgrass meadows had a higher diversity of SOB. Our result indicated that there were regional differences in the system that detoxifies H_2_S in eelgrass meadows, either microbial oxidation mediated by SOB or O_2_ permeation via the physical diffusion of benthos. However, since bacterial flora and phylogenetic analyses cannot show bias and/or causality due to PCR, future kinetic studies on microbial metabolism are expected.

## 1. Introduction

*Zostera marina* also known as eelgrass is the most widely distributed marine flowering plant in the northern hemisphere. It is a key component of marine ecosystems, providing geological stabilization through the growth of stolon and enables marine organisms to spawn in the eelgrass meadows [1,2,3]. Eelgrass, with its high photosynthetic capacity, has been reported to absorb approximately 17% of the carbon from seawater and directly from the atmosphere when it is exposed to the atmosphere at low tide [4]. Eelgrass meadows in coastal areas are disappearing due to the loss of shallow areas caused by land reclamation, changes in ocean currents, and industrial wastewater with Japan’s economic growth [5]. In general, chemical composition analysis of the eelgrass meadows sediment has shown that eelgrass meadows have low dissolved oxygen concentrations and high concentrations of toxic reducing substances, such as iron, manganese, and sulfides [6]. Sulfides, such as H_2_S, are typically toxic to eukaryotic cells even at concentrations as low as 1–10 mmol/L. Eelgrass has been reported to be sulfide tolerant at concentrations up to 4 mmol/L in situ and in vitro as it leaches some of the oxygen produced through photosynthesis from the root tips [7,8]. On the other hand, H_2_S above 4 mmol/L is toxic to eelgrass, and excessive accumulation can lead to the decimation of eelgrass meadows. Magnificent and sustainable eelgrass meadows require not only nutrient sources but also sediment conditions that provide an H_2_S balance. Recent reports have shown that healthy sediment microflora supports huge eelgrass meadows and that each has its own unique characteristics [9]. For example, microorganisms involved in the sulfur cycle, such as sulfate reduction and sulfide oxidation, have been detected in the microflora of the eelgrass meadows sediments. The regional nature of these microflora is attributed not only to ocean currents and coastal development but also to human life and ethnic cultures. However, there is limited research that has investigated the relationship between microflora and chemical components, such as sulfide, including regional characteristics [10,11]. Therefore, it is important for the future conservation of eelgrass meadows to determine, from comprehensive surveys, the factors that are essential for maintaining the expansion of eelgrass meadows. In this study, the eelgrass meadows of different sizes in Japan were systematically characterized by analyzing the microbial communities and chemical components of four geographically distant eelgrass meadows along the Japanese coast.

## 2. Results

### 2.1. Microbial Composition in Eelgrass Meadows

The sequencing reads were obtained to fully reflect the alpha diversity based on the Shannon index. Appendix AA shows a box plot for alpha diversity based on the presence or absence of the *Zostera marina* habitat condition. Bacterial diversity in the sediments did not differ dominantly between the eelgrass meadows and bare sediment, with a *p*-value of 0.77. The greatest diversity was found in eelgrass at the Wakayama coast, and the lowest diversity was found in eelgrass at the Kanagawa coast (Appendix A). On the other hand, the nonmetric multidimensional scaling method based on the Bray–Curtis distance matrix indicated that the eelgrass meadows sediments and bare sediments in Kumamoto clustered significantly apart from the other three sites. Kanagawa and Kumamoto, where there are larger eelgrass meadows, showed a lesser difference in diversity compared to bare sediment, while Osaka and Wakayama, where the eelgrass is smaller meadows, were clearly plotted far apart from the eelgrass meadows and bare ground.

Figure 1 shows the results of the 16S metagenomic analysis based on the order level in eelgrass and bare sediment at each collection site. Each color in the bar graph indicates a bacterial species at the order level. The phyla that contributed to less than 1% of the total abundance were combined and referred to as “Others”. *Bacteroidales*, *Campylobacterales*, *Chromatiales*, and *Thiotrichales*, as typical bacteria, were present in all eelgrass sediments at greater than 1%. The common bacterial orders of eelgrass and bare ground were *Anaerolineales*, *Babeliales*, *Cytophagales*, and *Phycisphaerales*, marine bacteria that were found to comprise more than 1% of all sediment and marine sediment. Sulfate-reducing bacteria (SRB), which contributed to sulfate reduction, were present at 18.40% (±7.14%) in eelgrass sediment compared to 6.66% (±3.50%) in bare sediment. This indicated that their presence was approximately twice as abundant in the eelgrass sediment than in the bare sediment. Sulfur-oxidizing bacteria (SOB) were present at 7.14% (±2.16%) and 4.74% (±1.37%) in the eelgrass and bare sediment, respectively. There was relatively no difference in the eelgrass sediment at any location, whereas there was no SOB detected in the bare sediment in the Wakayama and Osaka sediment samples.

Linear discriminant analysis using the Linear Discriminant Analysis Effect Size (LEFSe) showed that Chlomatiales (LDA score: 4.23, *p*-value: 0.02) were the dominant microbes in eelgrass sediment and *Chitinophagale* (LDA score: 4.05, *p*-value: 0.02) were the main microbes in bare sediment (Appendix A). The two governing bacterial orders were positioned distally in terms of phylogenetic relationships (Appendix A). Figure 2 shows a heatmap of the microbial composition based on an order level of more than 1% in each sediment. The dendrogram, based on the similarity of Euclidean distances at the order level, showed that the microbial communities of all sediments were divided into two groups (yellow: group 1 and green highlight: group 2). Group 1 consisted mainly of aerobic bacteria, such as *Gammaproteobacteria*, *Alphaproteobacteria*, *Betaproteobacteria*, and nitrifying bacteria, while group 2 consisted of anaerobic bacteria, such as SRB and SOB.

Sample clustering showed that the respective coastal bottom sediments were divided into two main clades, eelgrass meadows sediments and bare ground sediments, except in the Katakui coast, shown as Wakayama_E. Clade 1 consisted of bare ground at the Ozaki coast (Osaka) and Marine Park (Kanagawa) and eelgrass sediment on the Katakui coast. Clade 2 consisted of eelgrass sediment on the Ashikita coast (Kumamoto), Marine Park (Kanagawa), and Ozaki coast (Osaka). Sediments belonging to clade 1 showed a high prevalence of group 1 bacteria, which included rhizobia and nitrifying bacteria, and a low abundance of group 2 bacteria. The Katakui coast (Wakayama) eelgrass sediments showed similar bacterial abundances of both groups. In contrast, in clade 2, the composition of bare sediment was similar to that of the Ashikita (Kumamoto) eelgrass sediment. The Ashikita coast sediments contained the SRB *Thiotrichales* and *Syntrophobacterales*. The other types of SRB detected were Desulfobacterales and Desulfuromonadales in the Ozaki (Osaka) eelgrass sediment. However, no SRB were located in the marine park (Kanagawa) eelgrass sediments.

### 2.2. Determination of Total Bacteria and Chromatiales Population

The total number of bacteria and Chromatiales was calculated using qPCR (Appendix A). The total bacterial cells ranged from 3.5 × 10^5^ to 2.2 × 10^10^ cells/g-sediment and varied broadly between the different sampling sites, except for the Ashikita coast (Kumamoto) sediment, wherein the bare sediment was confirmed to be 10 to 1000 times more abundant than the eelgrass sediment. Chromatiales had the lowest population (5.2 × 10^3^ cells/g-sediment) in the Marine park (Kanagawa) sediment and the highest population (5.8 × 10^5^ cells/g-sediment) in the Ashikita coast (Kumamoto) sediment. Chromatiales accounted for approximately 1.2 × 10^−6^ to 8.4 × 10^−3^ of the total number of bacteria. The Chromatiales were approximately 4.7 times higher in the eelgrass meadows sediment than in bare sediments.

### 2.3. Determination of Chemical Components in Eelgrass Sediment

The DO, ORP, H_2_S, TOC, Fe, Ca^2+^, K^+^, and NO_3_^−^ components in each sediment sample are shown in Table 1. Low DO levels were observed in eelgrass sediment at all sites and were in good agreement with ORP behavior. This supports the hypothesis that an anaerobic environment is an important factor for eelgrass germination. However, H_2_S molecules are harmful to plants, such as eelgrass, despite existing in all eelgrass meadows sediments. The lowest level of H_2_S was 2.70 ppm (STD: 1.37) in the Marine park (Kanagawa), while the highest level of 23.29 ppm (STD: 4.34) was measured in Katakui coast (Wakayama). These H_2_S concentration levels at four sites were on average 11.8 times higher than those in the bare sediments. NO_3_^−^ was highly dependent on the sampling site and showed no correlation with the other chemical components.

Figure 3A shows that the eelgrass sediment in Marine Park (Kanagawa) had an overall balanced profile. The eelgrass sediment on the Ozaki coast (Osaka) showed higher-than-average values for NO_3_^−^, Fe, and K^+^. On the Katakui coast (Wakayama), Fe was lower than in the other sites and Ca^2+^ was higher than in the other sites. The Ashikita coast (Kumamoto) sediments had lower NO_3_^−^ and DO levels than those of the other sites. In contrast, the Marine Park (Kanagawa) bare sediment showed higher positive ORP values than the eelgrass sediment. The Ozaki coast (Osaka) bare sediment was characterized by NO_3_^−^, K^+^, and Fe values that were significantly lower than those of the eelgrass sediments and had positive ORP values. The Katakui coast (Wakayama) bare sediment showed higher DO values than eelgrass sediment. The Ashikita coast (Kumamoto) sediment showed similar profiles between the eelgrass and bare sediment.

Figure 4 indicates the relationship between the determined chemical components (ORP, DO, H_2_S, TOC, Fe, K^+^, Ca^2+^, and NO_3_^−^) and the principal component analysis. The plot shape indicates eelgrass vegetation conditions, and the color indicates the sampling site. The arrows reflect the orientation of the variables used in the analysis relative to the principal components, and the length of the arrows reveals the strength of the correlations with each principal component, which in the PCA1 and PCA2 planar plots reflect approximately 71% of the total data. An evaluation of the overall distribution of the plots showed that the bare sediments formed small clusters at all three sites, except at the Ashikita coast (Kumamoto), and eelgrass sediments formed clusters in all the regions. Overall, the first principal component explained 57.1% of the variance, correlated negatively with DO and ORP, and correlated positively with H_2_S, TOC, Ca^2+^, K^+^, and Fe. The second principal component explained 14.5% of the variance with a strong negative correlation with NO_3_^−^. Therefore, these components are reflected as the anaerobicity and nitrogen source content of the seafloor sediments, respectively. In the first quadrant, eelgrass meadows and bare sediments mainly from the Ashikita coast (Kumamoto) were plotted and correlated positively with the organic matter content. The H_2_S, SOB, and eelgrass sediments from the Katakui coast (Wakayama) also correlated slightly with SOB. In the second quadrant, eelgrass and bare sediments mainly from Marine Park (Kanagawa) were plotted closely to the bare sediments on the Katakui coast (Wakayama). Eelgrass sediments from the Ozaki coast (Osaka) were plotted in the fourth quadrant, and bare sediments were plotted in the fourth quadrant. From these PCA plots, the eelgrass sediments formed a separate cluster from the bare sediments; however, within that cluster, they formed small sub clusters by region. In contrast, the bare sediments were found to have similar bottom sediment conditions with no regional differences.

## 3. Discussion

### 3.1. Characteristics of Bacterial Composition in Eelgrass Sediments

This study provides insights into the microbial composition and chemical profiling in geographically distant eelgrass meadows along the Pacific coast in Japan. While the microbial composition and chemical composition of local eelgrass meadows have been analyzed regionally in a previous study, it has become evident that the microbial and chemical profiles in sediment vary depending on the scale and coast region [3,9,12]. Eelgrass meadows creation is an important coastal environment for biodiversity and a decarbonized society. In recent years, eelgrass meadows have featured prominence not only as an essential factor in marine ecosystems, contributing to stabilization, but also as a significant player in the realm of blue carbon. However, eelgrass meadows creation has sometimes been difficult to scale up and sustain. The objective of this study was to comprehensively investigate the bacterial and chemical composition of eelgrass sediments in geographically distinct coastal areas of Japan in order to identify similarities and the regional characteristics of eelgrass sediments for the sustainable creation of eelgrass meadows. Previous studies surveyed the Japanese coastal areas using a combination of bacterial composition analysis and chemical composition determination [13,14,15]. In another report for the Japanese coast, a multifaceted investigation was conducted via a geographical and biochemical assessment to “visualize” a causal structural network of symbiotic bacterial communities as photosynthetic bacteria living in the bottom mud of thriving eelgrass [9]. Furthermore, since eelgrass actively releases photosynthetic products and absorbs nutrients from marine sediment through its stolon, eelgrass survival is strongly related to factors in the rhizosphere. Therefore, the bacterial composition in the rhizosphere of geographically separated eelgrass and bare sediments was evaluated in this study. These bacterial compositions in eelgrass sediment indicated similarities to previous reports, which were investigated worldwide. In particular, SRB (*Desulfobulbaceae*, *Desulfovibrionaceae*, *Desulfuromonadaceae*, or *Desulfobacteraceae* families) and SOB (*Sulfurimonas* family) were detected in eelgrass meadows along the west and east coasts of the United States, Yellow Sea, Baltic Sea, Irish Sea, and Mediterranean Sea, which showed good agreement with our results. Therefore, these SRBs and SOBs are characteristic bacteria in eelgrass meadow sediments [12,16]. As the sediment depth increases, the concentration of dissolved oxygen typically decreases. In anaerobic environments, SRB proliferate using the sulfate abundant in seawater as a substrate producing H_2_S. In marine sediment thus enriched in H_2_S, general aerobic bacteria are weeded out [6]. The SRB were approximately twice as abundant in the eelgrass sediment as in the bare sediment (Figure 1), suggesting that H_2_S production via sulfate reduction may be more active in eelgrass sediment than in the bare sediment as previously reported and also in the sites covered in this study. Sediments from eelgrass and bare sediment on the Ashikita coast (Kumamoto) showed very similar bacterial compositions. This was due to the homogenization of marine sediments caused by the inflow of sediment from the mountains as a result of the landslide that occurred in July 2020. A heat map, used to examine the bacterial flora in more detail, revealed the proportion of each bacterial order. This indicated a high proportion of sulfate-reducing bacteria as a common feature in eelgrass sediments (Figure 2). The eelgrass sediments in Marine Park (Kanagawa) contained below average percentages of SRB, *Desulfobacterales*, and *Desulfuromonadales* on the Ozaki coast (Osaka), *Thiotrichales* and *Desulfuromonadales* on the Katakui coast (Wakayama), and *Thiotrichales* and *Syntrophomonadales* in Ashikita (Kumamoto). *Thiotrichales* and *Syntrophobacterales* were the main SRB. The SRB were more abundant in eelgrass sediments than in bare sediments; however a different order of SRB was detected, suggesting that the types of SRB present reflect the characteristics of eelgrass sediments in each region. In contrast, the presence of SOB, which oxidizes hydrogen sulfide to sulfate ions or single sulfur, was 2.6 times higher in eelgrass sediments than in bare sediments. The main SOB were as follows: *Thiomicrospirales*, *Flavobacteriales*, *Ectothiorhodospirales*, and *Chromatiales* in the Marine Park (Kanagawa), *Ectothiorhodospirales* and *Chromatiales* on the Katakui coast (Wakayama), *Flavobacteriales* on the Ashikita coast (Kumamoto), and *Ectothiorhodospirales*, *Chromatiales*, and *Flavobacteriales* on the Ozaki coast (Osaka). The composition of these bacterial flora was in some respects consistent with the report by Miyamoto et al. introduced earlier [9]. However, our results clearly indicate a marine-specific bacterial flora that is characteristic of the region, supporting the regional nature of eelgrass meadow sediments.

### 3.2. Detoxification System in Eelgrass Sediments

In general, H_2_S is reported to inhibit the initial growth and seeds maturing for plants under anaerobic conditions in waterlogged sediment. In particular, the eelgrass meadow sediment on the seafloor is a reducing environment with low oxygen and nutrients, and thus, H_2_S accumulation by SRB is likely to occur [17,18]. According to previous studies, hydrogen sulfide in marine sediments is toxic to eelgrass at concentrations over 4 mM [8]. Therefore, it is necessary to detoxify the irregularly produced hydrogen sulfide to maintain eelgrass sediment. Thus, in terms of microbial H_2_S oxidation, the higher the diversity of SOB, the more efficient the detoxification of H_2_S. The *Chromatiales* bacterial population, as a representative SOB, was estimated via the absolute quantification method using qPCR (Figure 3). Marine park (Kanagawa) had the highest SOB diversity among all the sites, and the total bacterial population was higher in eelgrass sediment than in bare sediment. Figure 5 summarizes the H_2_S formation and detoxification pathways in the eelgrass meadows ecosystem. The H_2_S is formed under anaerobic conditions via the sulfate reduction of SRB from SO_4_^2−^ and detritus, which are abundant in seawater and sediment. The detoxification of H_2_S is performed by the following four pathways: firstly by the microbial oxidation in the eelgrass rhizosphere by the SOB; secondly is chemical oxidation by the oxygen generated from the eelgrass roots; thirdly is the chemical oxidation by NO_3_^−^ present in the seawater; and finally is the chemical oxidation by Fe and its compounds. This ecosystem is similar to the results of rice meadows improvement studies that have been reported for many years. Since H_2_S accumulated in rice meadows causes reduced nutrient uptake and various diseases in rice plants, there are examples of countermeasures to precipitate and remove H_2_S by Fe_2_O_3_ [19,20,21]. It is also reported that NO_3_^−^ acts as an oxidant to oxidize H_2_S in marine sediments where H_2_S accumulates in a previous study [22]. Our results indicated that each eelgrass meadow had a different detoxification system (Figure 4 and Figure 5). There was a detoxification system with NO_3_^−^ along the Ozaki coast (Osaka), indicative of an urban coastal environment. The Katakui (Wakayama) and Ashikita coasts (Kumamoto) were significantly affected by the SOB detoxification systems. These coastal areas are close to mountainous areas and have relatively high inflow from the mountains. The inflow of spring water and sediment from mountains is linked to the natural supply of minerals and microorganisms. However, the Marine Park (Kanagawa) showed a unique environment. Dissolved oxygen was supplied to the bottom sediment of the eelgrass meadows that had established an anaerobic environment. Despite the relatively large size of the eelgrass meadows, the system that enabled an extensive supply of oxygen can be attributed to the activities of Benthos. Benthos, such as bivalves and lugworms, are more abundant in the bottom sediment of the eelgrass meadows in Marine Park (Kanagawa) than in other eelgrass meadows, and the physical agitation of the bottom sediment is thought to be responsible for the detoxification of H_2_S by delivering oxygen to the reduced bottom sediment. We suggest that knowledge of the sediment factors affecting healthy eelgrass meadows will enable the selection and implementation of H_2_S detoxification systems that are appropriate for the site.

## 4. Materials and Methods

### 4.1. Sampling Sites and Sample Collection

Eelgrass and bare sediments were collected from four coastal sites (Figure 6): Marine Park Kanagawa Prefecture (35.3378 N, 139.6370 E), Katakui coast Wakayama Prefecture (33.9398 N, 135.0850 E), Ozaki coast Osaka Prefecture (34.3727 N, 135.2455 E), and Ashikita coast Kumamoto Prefecture (32.2994 N, 130.4761 E). The large eelgrass meadows were located at Marine Park (Kanagawa) and the Ashikita coast (Kumamoto) and measured 77,804 (600 m long) and 44,200 m^2^ (700 m long), respectively. Katakui (165 m^2^ and 5 m long at Wakayama) and Ozaki coasts (1200 m^2^ and 10 m long in Osaka) were dotted with patchy small eelgrass meadows. Each sediment sample was collected at a 6 cm depth or less from the sea bottom through coring (6 cm in inner diameter, 20 cm in height, polyvinyl chloride tube), and the oxidation reduction potential (ORP) and dissolved oxygen (DO) were measured at the coring site. Each sediment sample was immediately ice-cooled in a cooler and frozen in a −5 °C freezer within 3 h and then transported frozen to the laboratory. Approximately 30 g of the frozen sediment samples was transferred to sterile 50 mL plastic tubes and stored in a −20 °C until DNA extraction and metagenomic amplicon sequencing.

### 4.2. 16S Metagenomic Sequencing

Genomic DNA from the collected sediment was extracted using the ISOIL for Beads Beating kit (NIPPON GENE Co., Tokyo, Japan). Extracted DNA was amplified using the 515F (5′-GTGCCAGCMGCCGCGGTAA-3′) and 806R (5′-GGACTACHVGGGTWTCTAAT-3′) primers targeting the v4 hypervariable region of the 16S rRNA gene [23,24]. Gene RED PCR Mix Plus was used for amplification under the following conditions: 95 °C for 5 min; 30 cycles of 95 °C for 20 s, 65 °C for 20 s, 72 °C for 5 s; and 72 °C for 10 min. Agarose gel electrophoresis confirmed the presence of DNA, and DNA libraries were constructed using the MiSeq Reagent Kit V2 (500 cycles) (Illumina Co., San Diego, CA, USA). The resultant libraries were quantified using a dsDNA HS Assay Kit on a Qubit 2.0 Fluorometer (Thermo Fisher Scientific Co., Waltham, MA, USA). The DNA library was sequenced as paired-end sequences with an insert size of 380 bp using the Illumina MiSeq (Illumina Co., San Diego, CA, USA).

### 4.3. Microbial Community Composition and Diversity Analysis

Raw reads were processed using the Qiime2 software (ver. 2021.8). The DADA2 algorithm was used for reducing noise and for trimming the low-quality regions of the raw reads. The acceptable or passed reads were used for further analysis [25]. The sequences were clustered into operational taxonomic units (OTUs) with 99% sequence similarity using VSEARCH (ver. 2.7.0), and chimeras were removed with the USEARCH software (ver. 6.1.544) [26]. Taxonomic OTUs were linked by mapping the 16S rRNA gene amplicons to the Silva database (ver. 138), which was used as the reference database. LEFSe (Python ver. 3.9.10) was performed with bacterial abundance at the order level to identify bacterial species significantly present in eelgrass sediment and bare sediment. LEFSe was performed using the parameters with a *p*-value of ≤ 0.05 and LDA (linear discriminant analysis) score values of 3.0 and below (python ver. 3.9.10). To investigate the differences between eelgrass and bare conditions in the b diversity of the microbial community composition, the PERMAMOVA(permutational multivariate analysis of variance) test was estimated by calculating the Bray–Curtis, which was based on the relative abundance [27].

### 4.4. qPCR Analysis for Total Bacteria and Sulfur-Oxidizing Bacteria Population in Sediments

qPCR assays were performed using a Rotor-Gene Q (QIAGEN Co., Hilden, Germany) under the following reaction conditions: 12.5 μL QuantiTect SYBR Green PCR Master Mix (QIAGEN Co., Hilden, Germany), 1 μL of each primer, and extracted DNA was added to a final volume of 25 μL. Total bacteria and sulfur-oxidizing bacteria in each sediment were determined via quantitative PCR using the extracted DNA, according to the manufacturer’s instructions. The quality of the extracted DNA in each sample was verified using a Qubit2.0 Fluorometer (Thermo Fisher Scientific Co., Waltham, MA, USA). The 16S rRNA gene for total bacteria and the specific 16S rRNA gene for *Chromatiales*, as sulfur-oxidizing bacteria, were amplified according to the thermocycling conditions described in Table 2. The 16S rRNA gene was amplified using the universal bacterial primers Eub519F (5′-CAGCAGCCGCGGTRATA-3′) and U785R (5′-GGACTACCVGGGTATCTAAKCC-3′). The specific 16S rRNA gene for *Chromatiales* was amplified using the primers CHR986F (5′-AGCCCTTGACATCCTCGGAA-3′) and CHR1392R (5′-ACGGGCGGTGTGTAC-3′) [28]. The extracted 16S rRNA, for the calibration curve, from *Escherichia coli* DH5α (JCM1649^T^) and *Thiocapsa subramanianensis* (JCM14886^T^) were cultured in LB media and PFENNIG’s medium, respectively. The SYBR green staining method was used to determine the number of cells in the culture media. Respective calibration curves were prepared using DNA extracted from each culture. Absolute quantification was performed using Rotor-GeneTM Q Software (ver. 2.0.2) based on the ”Crossing Point” value that defines the cycle number at which the fluorescence signal of the sample exceeds a background fluorescence value [29,30]. The Ct value was calculated as the intersection of the generally used threshold value and the amplification curve in the software.

### 4.5. Chemical Component Analysis

In all collected sediments, the geochemical indicators, such as ORP, DO, H_2_S, Fe, K^+^, Ca^2+^, and NO_3_^−^, were determined. The ORP and DO were measured in 6 cm of sediment at each sampling site. The ORP was measured with a portable water quality meter, LAQUA D-210C and D-210PC (HORIBA Co., Kyoto, Japan), and the DO was measured using a digital dissolved oxygen meter PDO-520 (FUSO Co., Kanagawa, Japan). Spectrophotometric analysis was used to determine the H_2_S concentrations using the methylene blue method with 670 nm [31].

In the Fe content determination, 1 g of each sediment was treated with 4.5 mL of 0.5 M hydrochloric acid for a 24 h treatment at room temperature to extract Fe ions [31,32]. Subsequently, the Fe content in all sediments was measured with an atomic absorption spectrometer (Z-2300, HITACHI Co., Tokyo, Japan). The K^+^ and Ca^2+^ were determined with an ion chromatography method using an HPLC system (LC-20AP system, SHIMADZU Co., Kyoto, Japan) equipped with a Shim-pack IC-C4 column (150 mm 4.6 mm, SHIMADZU Co., Kyoto, Japan). All sediments were washed with distilled water and filtered through a 0.45 mm membrane filter. The eluent was 3 mM (COOH)_2_, and detection was performed based on electrical conductivity. After treating all the specimens according to the brucine method, the NO_3_^−^ concentration was determined at a wavelength of 410 nm using the absolute calibration curve method [33].

### 4.6. Total Organic Component Analysis

Thermogravimetric analysis was conducted to measure the organic content of the marine sediment. Each sediment was heated at 100 °C to remove moisture and increased to 600 °C to vaporize organic contents. The organic matter content was calculated and compared with the sediment weight obtained at 100 and 600 °C. The mass lost after heating can be considered as the total mass of the organic content. Total organic content analyses were performed on all collected sediments [34].

### 4.7. Statistical Analysis and Data Visualization

Statistical analysis and data visualizations were conducted using the R studio software (ver. 2021.09.0) with the biplot and ggplot2 packages. To capture the characteristics of all sediments, PCA plots was visualized with physicochemical measurements, and the estimated sulfur-oxidizing bacteria counts, as parameters collected from eelgrass meadows and bare sediment at four sites, visualized correlations regarding H_2_S detoxification by persistent eelgrass meadows.

## 5. Conclusions

The conservation of eelgrass meadows has been focused in terms of carbon neutral, climate change, and marine biodiversity. This report showed the importance of H_2_S detoxification systems to create a sustainable eelgrass environment. Interestingly, the statistical analysis results and field observations suggest that benthic organisms, SOB, and DO contribute to the presence of eelgrass meadows and that they are involved in the removal of H_2_S accumulated in the bottom sediment. However, since bacterial flora and chemical analyses of bottom sediments cannot show the bias level of PCR amplification and the chemical species involved in microbial metabolism, future interpretation is expected to be based on dynamic experiments using a metatranscriptome analysis and other methods.

## Figures and Tables

**Figure 1 ijms-25-05442-f001:**
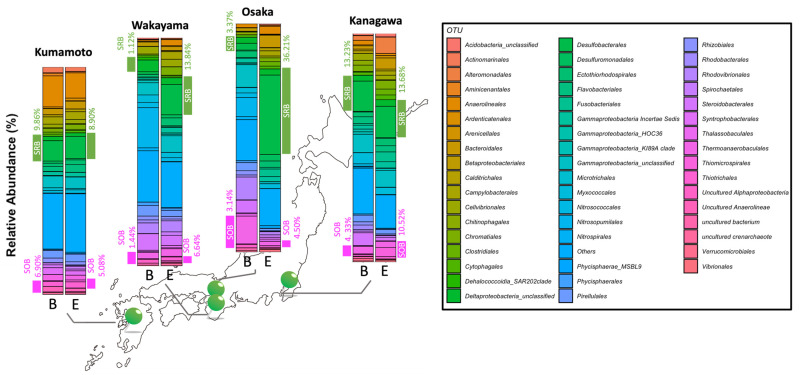
Barplot of microbial community compositions in each eelgrass and bare site based on the taxonomic order level at eelgrass sites. E and B show eelgrass and bare meadows, respectively. Taxonomy of the representative sequences with the Qiime2 and SILVA database (ver. 138 database clustered to 99% OTUs). Others shows the orders that contributed less than 1% in each sediment.

**Figure 2 ijms-25-05442-f002:**
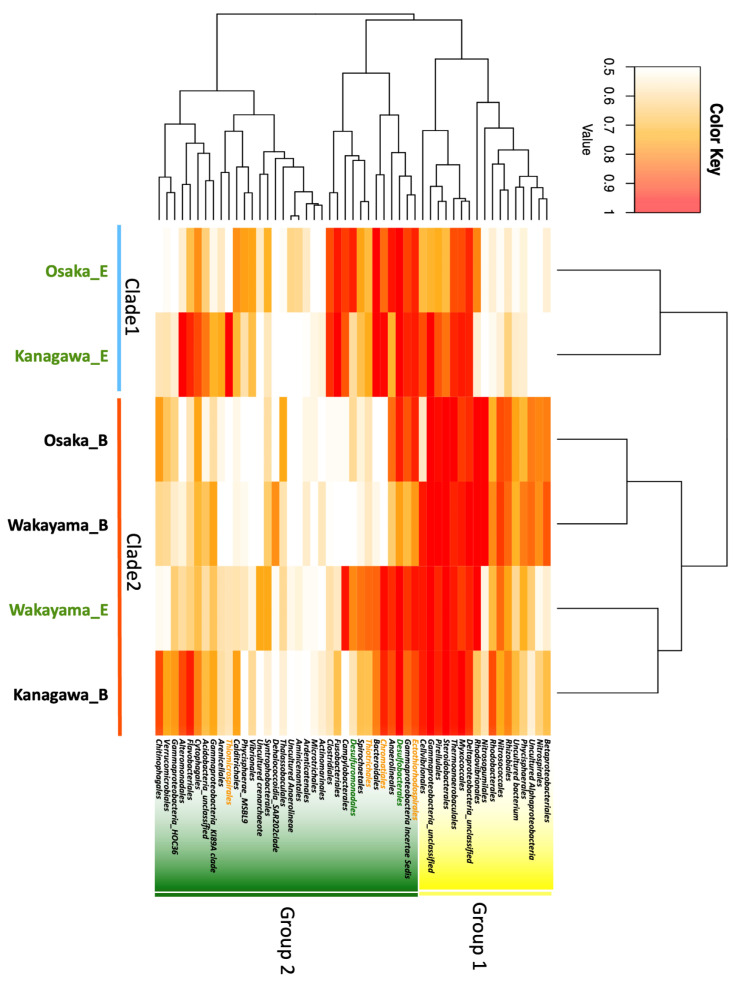
Heatmap of clustering analysis of the relationship between sediment and the taxonomic order. The color code indicates the abundance of organisms and ranges from white (low abundance) to red (high abundance).

**Figure 3 ijms-25-05442-f003:**
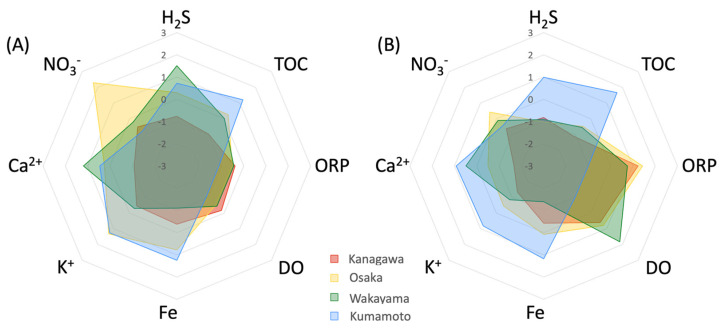
Radar chart of all chemical indicator (ORP, DO, H_2_S, TOC, Fe, K^+^, Ca^2+^, and NO_3_^−^) value as Z-scores. (**A**) Eelgrass meadows sediment and (**B**) bare meadows sediment.

**Figure 4 ijms-25-05442-f004:**
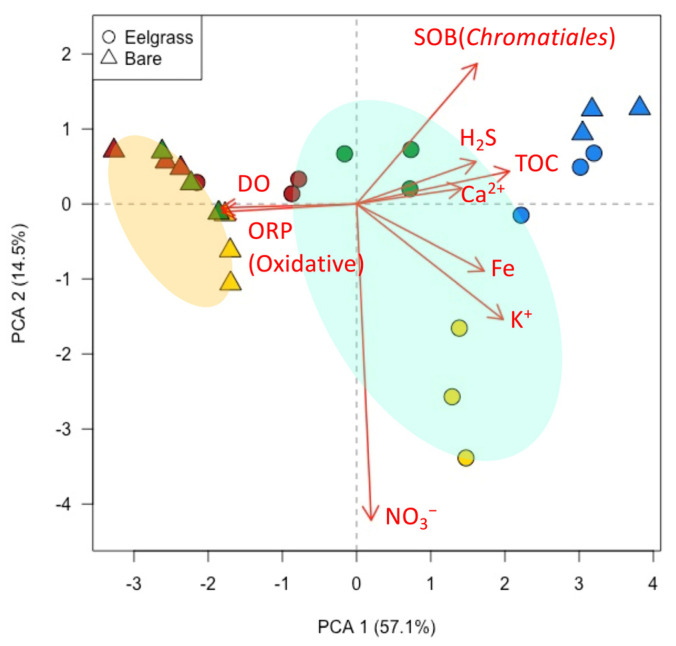
PCA plot of chemical and bacterial profiling in each sediment. Circle plots indicate eelgrass meadows and triangle plots indicate bare sediments. Red, yellow, green, and blue show Marine Park Kanagawa, Ozaki coast Osaka, Katakui coast Wakayama, and Ashikita coast Kumamoto Prefecture, respectively.

**Figure 5 ijms-25-05442-f005:**
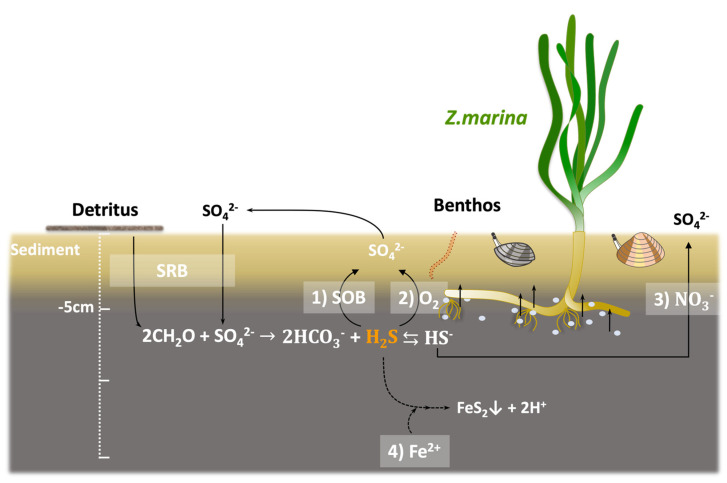
Sulfate reduction and sulfide cycle in eelgrass rhizosphere with biological, botanical, and chemical detoxification. Detoxification shows as follows: (1) microbial sulfide by SOB, (2) photo-oxidation by root discharge of O_2_ produced by the eelgrass’s own photosynthesis, and (3) direct chemical oxidation by iron ions and iron compounds.

**Figure 6 ijms-25-05442-f006:**
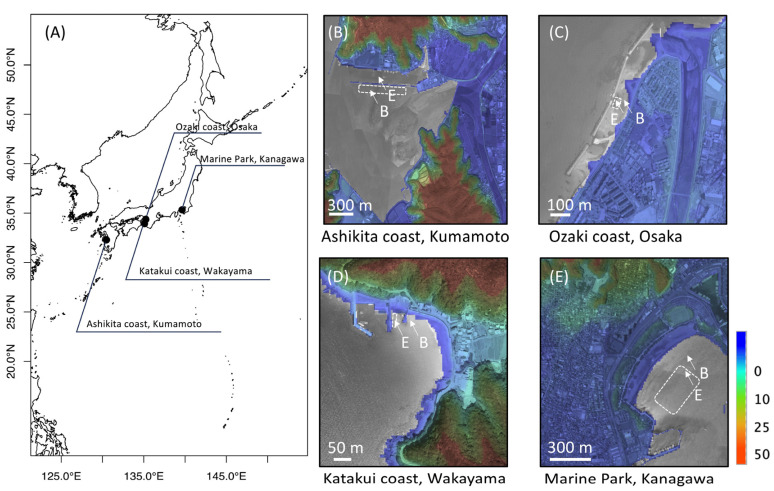
Eelgrass meadow map of sample collection site. The dashed square shows the eelgrass meadows field. (**A**) Wide map of the collection area. (**B**–**E**) Local maps of E: eelgrass meadow and B: bare meadow. The color scale on the right represents elevation according to the elevation data of the Geospatial Information Authority of Japan.

**Table 1 ijms-25-05442-t001:** Concentration of chemical indicators (ORP, DO, H_2_S, TOC, Fe, K^+^, Ca^2+^, and NO_3_^−^) for each sediment (*n* = 3). Values in parentheses indicate standard deviations.

		ORP/mV	DO/ppm	H_2_S/ppm	TOC/%	Fe/ppm	K^+^/ppm	Ca^2+^/ppm	NO_3_^−^/ppm
Eelgrass	Kanagawa	−154.5 (201.39)	3.0 (2.3)	2.70 (1.37)	1.28 (0.13)	393.0 (19.7)	36.7 (2.7)	65.0 (5.3)	0.22 (0.023)
	Osaka	−175.0 (63.25)	1.9 (0.9)	12.32 (6.68)	2.76 (0.42)	671.4 (57.5)	58.6 (0.9)	108.4 (14.7)	95.10 (23.92)
	Wakayama	−163.6 (54.50)	2.1 (0.9)	23.29 (4.34)	2.30 (0.14)	214.7 (25.4)	38.6 (0.7)	138.9 (6.3)	19.67 (9.40)
	Kumamoto	−282.3 (6.41)	0.1 (0)	16.14 (0.20)	4.52 (0.18)	798.8 (20.1)	57.8 (0.7)	115.1 (1.4)	0.031 (0.025)
Bare	Kanagawa	131.6 (68.69)	5.3 (1.0)	2.24 (1.57)	1.20 (0.09)	353.7 (12.6)	25.8 (4.0)	46.1 (9.3)	0.14 (0.022)
	Osaka	166.3 (1.00)	5.9 (0.7)	0.42 (0.21)	1.36 (0.33)	508.6 (28.8)	36.7 (2.7)	84.0 (4.4)	40.49 (12.49)
	Wakayama	46.2 (62.65)	9.1 (2.0)	1.47 (1.45)	1.23 (0.57)	144.0 (68.5)	31.8 (2.8)	116.4 (11.9)	24.19 (13.58)
	Kumamoto	−261.8 (8.49)	0.4 (0.2)	18.58 (9.96)	3.98 (0.25)	777.4 (25.8)	52.5 (1.1)	130.3 (4.3)	0.28 (0.40)

**Table 2 ijms-25-05442-t002:** qPCR programs for bacterial cell counts.

Primer Name	Target Organisms	Denaturation	Annealing	Extension
Eub519F/U785R	Total bacteria	5 s at 95 °C	30 s at 60 °C	30 s at 60 °C
CHR986F/CHR1392R	*Chromatiales*	30 s at 95 °C	60 s at 55 °C	120 s at 72 °C

## Data Availability

The 16S metagenomic sequences for each eelgrass sediment were deposited in DDBJ under BioProject number PRJDB14314, (https://ddbj.nig.ac.jp/resource/bioproject/PRJDB14314, accessed on 7 September 2022), BioSample number SAMD00533656-SAMD00533663 (https://ddbj.nig.ac.jp/resource/biosample/SAMD00533656, accessed on 7 September 2022), and accession number DRR403318-DRR403325 (https://ddbj.nig.ac.jp/resource/sra-run/DRR403318, accessed on 7 September 2022).

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
