# Peer review of "Microbial Detoxification of Sediments Underpins Persistence of Zostera marina Meadows"

_ijms, 2024, doi:10.3390/ijms25105442_

Round 1

Reviewer 1 Report

Comments and Suggestions for Authors

The study presented in the manuscript titled Microbial detoxification of sediments underpins persistence of  Zostera marina meadows examined bacterial and chemical composition data from eelgrass sediments along Japan's coast to understand their role in eelgrass acclimatization. It found that sulfur-oxidizing bacteria play a significant role in detoxifying hydrogen sulfide (H2S) in eelgrass meadows, with larger-scale meadows exhibiting greater diversity of these bacteria. This highlights the importance of sustainable conservation efforts for eelgrass beds. I recommend this paper for publication after minor revisions.

General comments:

The paper reads easily, introduction, results and discussion are well presented. Materials and methods and conclusion require few minor modifications. The quality of some figures is a bit low which makes the interpretation a bit difficult.

Specific comments:

Line 21: “investigate the sediment effect” “sediment” is not necessary.

Line 35: “Zoostera marina called” maybe “Zoostera marina also known as” sounds better.

Figure 2: Consider enlarging figure 2. Is impossible to read the right column. Additionally, what green and yellow refer to?

Table 1: Please, edit the table to have both the brackets on the same line, as in column ORP/mV.

Line 185: “they formed small sub cluster”

Line 195: “This study provides insights into the microbial composition and chemical profiling in geographically distant eelgrass meadows along the Pacific coast” of which country?

Line 198: Reference stile is different from the rest of the paper. Please, homogenize it.

Line 286: the title “Figure 5” should be in italic.

Line 307: “Figure 6” should be written in italic. Picture is blurry, better quality is required.

Line 312: (Nippon GENE Co.) City should be added after the name of the company. Please add this information for all the companies you mention in the text.

Line 314: is the reference for the primer the same as the one for the PCR condition?

Line 340: missing reference.

Line 350: Reference stile is different from the rest of the paper. Please, homogenize it.

Line 354: Please, briefly describe the method.

Line 355: What was boiled for 15 minutes? How you constructed your calibration curves?

Line 367: Please, briefly describe the method.

Line 390: conclusion should be amplified a bit. Why the detoxification activity of the eelgrass meadows is important? Why the coexistence of too many organisms can be problematic? How your research provides important insights for the scientific community?

Author Response

Thank you for your careful peer review. All remarks were edited together with the co-authors.

Line 21: “investigate the sediment effect” “sediment” is not necessary.

[response] We have reflected reviewer’s comment.

Line 35: “Zostera marina called” maybe “Zoostera marina also known as” sounds better.

[response] We have reflected reviewer’s comment.

Figure 2: Consider enlarging figure 2. Is impossible to read the right column. Additionally, what green and yellow refer to?

[response] We resized figure and right colummn. Green and yellow highlight is already explained in Line105-107.

Table 1: Please, edit the table to have both the brackets on the same line, as in column ORP/mV.

[response] We have reflected reviewer’s comment.

Line 185: “they formed small sub cluster”

[response] We have reflected reviewer’s comment. 

Line 195: “This study provides insights into the microbial composition and chemical profiling in geographically distant eelgrass meadows along the Pacific coast” of which country?

[response] Yes, that’s mean in Japan. We have reviced this sentence in L202.

Line 198: Reference stile is different from the rest of the paper. Please, homogenize it.

[response] We have reflected reviewer’s comment in L205.

Line 286: the title “Figure 5” should be in italic.

[response] We have reflected reviewer’s comment. 

Line 307: “Figure 6” should be written in italic. Picture is blurry, better quality is required.

[response] We have reflected reviewer’s comment. 

Line 312: (Nippon GENE Co.) City should be added after the name of the company. Please add this information for all the companies you mention in the text.

[response] We have reflected reviewer’s comment in L326.

Line 314: is the reference for the primer the same as the one for the PCR condition?

[response] Primer and  PCR condition are referenced in same reference. We were tried amplification at same primer and condition in L329.

Line 340: missing reference.

[response] Reference removed due to manufacturer's recommended protocol.

Line 350: Reference stile is different from the rest of the paper. Please, homogenize it.

[response] We have reflected reviewer’s comment in L365.

Line 354: Please, briefly describe the method.

[response] We have reflected reviewer’s comment. We added more detailed information in L365-368.

Line 355: What was boiled for 15 minutes? How you constructed your calibration curves?

[response] We revised this sentence in L370.

Line 367: Please, briefly describe the method.

[response] We have reflected reviewer’s comment in L384-387.

Line 390: conclusion should be amplified a bit. Why the detoxification activity of the eelgrass meadows is important? Why the coexistence of too many organisms can be problematic? How your research provides important insights for the scientific community?

[response] This section has received input from other reviewers, and we have edited the text to reflect all of them, L411-419.

Reviewer 2 Report

Comments and Suggestions for Authors

Dear authors,

I read your article with interest, which contains valuable information and required a large amount of work. However, the manuscript contains several shortcomings and requires additional qualifications so that it can be understood more easily by readers.

You have my observations below.

Abstract section. Lines 21-23. The authors give a certain percentage of the bacterial populations detected by molecular methods. Can their in situ activity be equated with the abundance estimated by PCR? Also, the percentage of SOB in a higher percentage in eelgrass does not necessarily certify a higher activity in this habitat compared to the sediments without eelgrass. Discriminant analysis can be a hint, but not a definite proof, that SOB detoxifies H2S. Bacteria can be present in a latent state or metabolically inactive from which DNA can be extracted and quantified. Therefore, I think it is necessary for the authors to establish the limits of the study right from the beginning of the article, respectively in the Abstract.

Lines 27-29. Which of the two factors, SOB or molecular oxygen are responsible to the greatest extent for H2S detoxification? The reaction between oxygen and H2S is a spontaneous chemical reaction, while the biological oxidation of H2S takes place under the action of specific bacterial enzymes, characteristic of SOB. Did the authors consider the detection of the activities of these enzymes? These questions and observations are also valid for the Results and Discussions sections.

Moreover, did the authors consider discriminating between chemical and biological activity?

Results section.

How do the authors explain the uniformity of bacterial diversity in eelgrass and the sediments lacking it? Figure 1, lines 69-70. In the figure there is a list of species without any specification of their percentage. What do the colors and shades next to the bacterial genera represent? Please enter the respective explanations so that the reader can understand the meaning of the figure better. Line 141. Please explain why an anaerobic environment favors the growth of eelgrass? Line 166. What is the main component of the analysis? Add additional explanations. Lines 175-177. Please explain what is the second main component of the analysis?

Discussion section.

Lines 199-200. I ask the authors to explain in detail how the eelgrass bed is for a decarbonized society. Lines 208-209. What intestinal bacteria are you talking about ("as photosynthetic bacteria and gut bacteria living in the bottom mud")? Eelgrass is a plant!!!

Lines 214-219. Please define the concept of the rhizosphere and its particularities in eelgrass. Lines 219-222. What are the causes of the increased abundance of SRB and SOB in the eelgrass bed? Lines 253-258. The detection by molecular methods of a bacterial group and their quantitative assessment does not always justify their biological activity in situ, respectively the reduction of sulfates or the oxidation of H2S. That is why, I think that the authors should emphasize from the beginning the limits of the study, namely that no testing of the oxidation rate of H2S by SOB was carried out, neither in the laboratory nor in situ. Lines 269-270. What evidence do the authors have that the H2S detoxification system uses NO3?

Materials and Methods section.

Line 352. I think the authors wanted to write Pfenning's medium. Please briefly describe the composition of the growing medium so that readers can better understand the context. Lines 350-352. Did the authors cultivate the "genes" or the organisms that contained those genes? Lines 367-369. Check the correctness of the sentence and please correct it for greater clarity. Lines 372-375. What followed the respective determination method? Lines 386-389. The formulation of the idea is not clear. Please rephrase.

Conclusions section.

Lines 390-398. The conclusions must be rewritten because there is no logical connection between the title of the article and the content of the conclusions. Did the authors study eelgrass germination and growth or microbial processes, as the title suggests? Lines 393-398. What is the meaning of the sentence? Lines 396-398. The phrase is not convincing regarding the role of bacteria in H2S detoxification. The authors should rethink and rewrite the Conclusions section. What are the original results of the study? What are the original contributions in relation to similar studies? And all this clearly expressed.

With best regards!

Comments on the Quality of English Language

English is good, only some typographical errors need to be corrected.

Author Response

Thank you for your precise comments. I have recapitulated the report of the research study at this time by clearly stating the purpose of this paper and the shortfalls.

Abstract section.
 Lines 21-23. The authors give a certain percentage of the bacterial populations detected by molecular methods. Can their in situ activity be equated with the abundance estimated by PCR? Also, the percentage of SOB in a higher percentage in eelgrass does not necessarily certify a higher activity in this habitat compared to the sediments without eelgrass. Discriminant analysis can be a hint, but not a definite proof, that SOB detoxifies H2S. Bacteria can be present in a latent state or metabolically inactive from which DNA can be extracted and quantified. Therefore, I think it is necessary for the authors to establish the limits of the study right from the beginning of the article, respectively in the Abstract.

[response] Sure, we show the limits of our study and this techniques at L27-30.

Lines 27-29. Which of the two factors, SOB or molecular oxygen are responsible to the greatest extent for H2S detoxification? The reaction between oxygen and H2S is a spontaneous chemical reaction, while the biological oxidation of H2S takes place under the action of specific bacterial enzymes, characteristic of SOB. Did the authors consider the detection of the activities of these enzymes? These questions and observations are also valid for the Results and Discussions sections.

Moreover, did the authors consider discriminating between chemical and biological activity?

[response] This line was misleading and has been corrected at L27-30.

Results section.

How do the authors explain the uniformity of bacterial diversity in eelgrass and the sediments lacking it? Figure 1, lines 69-70. In the figure there is a list of species without any specification of their percentage. What do the colors and shades next to the bacterial genera represent? Please enter the respective explanations so that the reader can understand the meaning of the figure better.

[response] Since this is the first part of the results section, the edited statistical graphs were shown, as well as the interpretation of the alpha diversity values. And we added new sentence at L68-71 and L80 about explanations the colors of bar plots.

Line 141. Please explain why an anaerobic environment favors the growth of eelgrass? Line 166. What is the main component of the analysis? Add additional explanations.

[response] We have reflected reviewer’s comment. We added the explanation that process of H2S genarated and accumulated mechanisms at L279-280.

Lines 175-177. Please explain what is the second main component of the analysis?

[response] We added component information at L182-183.

Discussion section.

Lines 199-200. I ask the authors to explain in detail how the eelgrass bed is for a decarbonized society.

[response] We added more detailed information about role of eelgrass at L207-209.

Lines 208-209. What intestinal bacteria are you talking about ("as photosynthetic bacteria and gut bacteria living in the bottom mud")? Eelgrass is a plant!!!

[response] I quoted only necessary information and renewed a sentence at L216-218.

Lines 214-219. Please define the concept of the rhizosphere and its particularities in eelgrass. Lines 219-222. What are the causes of the increased abundance of SRB and SOB in the eelgrass bed?

[response] We added more information at L229-233.

Lines 253-258. The detection by molecular methods of a bacterial group and their quantitative assessment does not always justify their biological activity in situ, respectively the reduction of sulfates or the oxidation of H2S. That is why, I think that the authors should emphasize from the beginning the limits of the study, namely that no testing of the oxidation rate of H2S by SOB was carried out, neither in the laboratory nor in situ. Lines 269-270. What evidence do the authors have that the H2S detoxification system uses NO3?

[response] We explained with new reference in L281-282.

Materials and Methods section.

Line 352. I think the authors wanted to write Pfenning's medium. Please briefly describe the composition of the growing medium so that readers can better understand the context.

[response] Sure, this is Pfenning’s medium. This medium composition was very complex and has been corrected to the official medium name in L369.

 Lines 350-352. Did the authors cultivate the "genes" or the organisms that contained those genes? Lines 367-369. Check the correctness of the sentence and please correct it for greater clarity. Lines 372-375. What followed the respective determination method? Lines 386-389. The formulation of the idea is not clear. Please rephrase.

[response] We have reflected reviewer’s comment. We added more detailed information at L367-370, L387-390 and L396-397.

Conclusions section.

Lines 390-398. The conclusions must be rewritten because there is no logical connection between the title of the article and the content of the conclusions. Did the authors study eelgrass germination and growth or microbial processes, as the title suggests? Lines 393-398. What is the meaning of the sentence?

[response] This section has received input from other reviewers, and we have edited the text to reflect all of them at L414-422.

Lines 396-398. The phrase is not convincing regarding the role of bacteria in H2S detoxification. The authors should rethink and rewrite the Conclusions section. What are the original results of the study? What are the original contributions in relation to similar studies? And all this clearly expressed.

[response] This section has received input from other reviewers, and we have edited the text to reflect all of them at L414-422.

Reviewer 3 Report

Comments and Suggestions for Authors

The manusciprt describes the role of microbial communities in the detoxification of sediments affecting eelgrass meadows, providing valuable insight into ecological management and conservation strategies.

In my opinion, the Authors should expand their discussion about the limitations and potential biases of these methodology used herein. For example, it encompasses discussions regarding primer specificity in qPCR as well as potential biases associated with metagenomic sequencing.

Also, I would like to know how the Authors would address the problem of the study being geographically confined only to Japanese coastal areas. How would expanding the research (in order to include diverse geographical locations) enhance the generalization of the findings.

Author Response

Thank you for your courteous comments and understanding of the full scope of the study. I will write my response below.

In my opinion, the Authors should expand their discussion about the limitations and potential biases of these methodology used herein. For example, it encompasses discussions regarding primer specificity in qPCR as well as potential biases associated with metagenomic sequencing.

[response] The limit of the present study was shown in the second half of the abstract.

Also, I would like to know how the Authors would address the problem of the study being geographically confined only to Japanese coastal areas. How would expanding the research (in order to include diverse geographical locations) enhance the generalization of the findings.

[response] Thank you for your comments regarding the potential of this study. We are currently planning to apply for a patent on this matter and would like to not specify it in this paper.

Round 2

Reviewer 2 Report

Comments and Suggestions for Authors

Dear authors,
Thank you for the answer and the effort made to improve the quality and clarity of the manuscript.

With best regards!

Author Response

Thank you for this accurate review.